# Shallow Landslide Susceptibility Models Based on Artificial Neural Networks Considering the Factor Selection Method and Various Non-Linear Activation Functions

**Deuk-Hwan Lee [1]** , **Yun-Tae Kim [2] and Seung-Rae Lee [1],***

[1] Department of Civil and Environmental Engineering, KAIST, Daejeon 34141, Korea; deukhwan@kaist.ac.kr
[2] Department of Ocean Engineering, Pukyong National University, Pusan 48513, Korea; yuntkim@pknu.ac.kr
* Correspondence: srlee@kaist.ac.kr; Tel.: +82-42-350-3617

**Abstract:** Landslide susceptibility mapping is well recognized as an essential element in supporting decision-making activities for preventing and mitigating landslide hazards as it provides information regarding locations where landslides are most likely to occur. The main purpose of this study is to produce a landslide susceptibility map of Mt. Umyeon in Korea using an artificial neural network (ANN) involving the factor selection method and various non-linear activation functions. A total of 151 historical landslide events and 20 predisposing factors consisting of Geographic Information System (GIS)-based morphological, hydrological, geological, and land cover datasets were constructed with a resolution of 5 x 5 m. The collected datasets were applied to information gain ratio analysis to confirm the predictive power and multicollinearity diagnosis to ensure the correlation of independence among the landslide predisposing factors. The best 11 predisposing factors that were selected in this study were randomly divided into a 70:30 ratio for training and validation datasets, which were used to produce ANN-based landslide susceptibility models. The ANN model used in this study had a multi-layer perceptron (MLP) structure consisting of an input layer, one hidden layer, and an output layer. In the output layer, the logistic sigmoid function was used to represent the result value within the range of 0 to 1, and six non-linear activation functions were used for the hidden layer. The performance of the landslide susceptibility models was evaluated using the receiver operating characteristic curve, Kappa index, and five statistical indices (sensitivity, specificity, accuracy, positive predictive value (PPV), negative predictive value (NPV)) with the training dataset. In addition, the landslide susceptibility models were validated using the aforementioned measures with the validation dataset and were compared using the Friedman test to check the significant differences among the six developed models. The optimal number of neurons was determined based on the aforementioned performance evaluation and validation results. Overall, the model with the best performance was the MLP model with the logistic sigmoid activation function in the output layer and the hyperbolic tangent sigmoid activation function with five neurons in the hidden layer. The validation results of the best model showed a sensitivity of 82.61%, specificity of 78.26%, accuracy of 80.43%, PPV of 79.17%, NPV of 81.82%, a Kappa index of 0.609, and AUC of 0.879. The results of this study highlight the effectiveness of selecting an optimal MLP model structure for shallow landslide susceptibility mapping using an appropriate predisposing factor section method.

**Keywords:** shallow landslide; GIS; artificial neural networks; multi-layer perceptron; activation function; factor selection; filter method; susceptibility mapping; Mt. Umyeon; Korea

## 1. Introduction

Shallow landslides are one of the most common and frequent geo-disasters that occur in mountainous regions [1]. In most areas of Korea, where approximately 63% of the territory consists of mountainous regions, soil layers are generally less than 2–3 m in thickness with underlying bedrock [2]. In addition, the annual rainfall in the central region of Korea is approximately 1200–1500 mm, and more than half of the annual precipitation is concentrated during the months from July to September due to the influence of the Monsoon season. Due to these topographical and climate conditions, Korean mountains are regarded as regions that are susceptible to shallow landslides [3,4]. According to the statistics of the Korea Forest Service from 1976 to 2018, an average of 34 casualties and 395 ha of landslides occur annually. Considering such figures, there is a growing national interest in the development of proactive technologies for the prevention and mitigation of landslide hazards.

Landslide susceptibility mapping is well recognized as an essential element in supporting decision-making activities for disaster prevention and mitigation, as it provides information regarding landslide-prone areas. However, reliable spatial prediction of landslides remains a challenging task due to its complexity as it is affected by various internal factors (e.g., hydro-geotechnical properties, lithology, forestry, geological structure, topographic conditions) and external factors (e.g., rainfall, the melting of snow, earthquakes, volcanic eruptions) [5,6]. To resolve these problems, many studies have been conducted over several decades with the goal of developing high-performance-based landslide susceptibility models through various approaches, which can be divided into two categories: physically based methods and data-driven methods.

Physically based methods of landslide prediction [7,8] are generally expressed as safety factors of slope stability, which refers to the ratio of soil shear strength to the shear stress of potential sliding surfaces in the slope. Such methods do not require a historical inventory of landslides when developing susceptibility maps but require detailed geotechnical properties and geometric conditions. As such, physically based models are more practical for site-specific areas with homogeneous conditions [9], as it is expensive and time-consuming to build up a database for applications in large-scale areas [10,11].

Data-driven methods of landslide prediction [10–12] estimate potential landslides by analyzing and interpreting the relationship between historical landslide data and various predisposing factors through the means of statistical or machine learning techniques without physical processes. Therefore, historical landslide data and various factors related to landslide occurrence should be collected as the first step for the landslide susceptibility mapping [13]. Recent advances in data mining and soft computing have made it possible to easily link with Geographic Information System (GIS) platforms, enabling landslide susceptibility assessment over wide areas [14].

According to the literature review, artificial neural network (ANN) models have been reported as a suitable machine learning method for predicting non-linear and complex phenomena [15,16]. Such models have been widely applied for landslide susceptibility modeling [17–20]. In a study of applying an ANN-based susceptibility model, Vasu et al. [21] improved the predictive ability of the ANN by integrating a hybrid feature selection and an extreme learning machine. Tien Bui et al. [22] compared two training algorithms (Levenberg–Marquardt and Bayesian regularization network) and found that the latter algorithm was more robust and efficient. Lee et al. [23] showed that an ANN model performed better with the weights of each factor being determined compared to without determining the weighting. Ermini et al. [24] compared two architectures of ANN models (Multi-Layer Perceptron and Probabilistic Neural Network) and obtained slightly better results with the former architecture. Despite these efforts, there is still a multitude of considerations that should be accounted for when developing an optimal ANN model capable of high levels of performance [25,26], such as factor selection, the number of neurons and layers, and activation functions.

In this study, landslide susceptibility maps of Mount Umyeon were produced using ANN models with consideration of various model architectures. The main objective of this study is to determine the optimal structure of the ANN model considering the factor selection method and various activation functions for high-performance-based landslide susceptibility mapping. In the factor selection stage,

information gain ratio and multicollinearity analysis were applied for the evaluation of predictive power and mutual exclusivity of the landslide predisposing factors. Once evaluated, the optimal architecture of the ANN model was selected with consideration of the number of neurons and various activation functions by evaluating model performance using receiver operating characteristics (ROCs), Kappa index, and various statistical evaluation measures. Finally, a non-parametric test (Friedman test) was conducted to compare the developed susceptibility models to confirm significant differences.

## 2. Study Area and Spatial Database

### 2.1. Description of the Study Area

The study area is Mount Umyeon, which is located in the southern part of Seoul Special City, Korea, between latitudes 37°27′00″N and 37°28′55″N and longitudes 126°59′02″E and 127°01′41″E, as shown in Figure 1. This area covers a surface area of approximately 5.1 km², with the highest elevation being 293 m above sea level. The geological setting of this area is mainly composed of Pre-Cambrian banded biotite gneiss and granitic gneiss. The annual average precipitation of this area is 1450 mm. Extreme heavy rainfall from 26 July to 27 July in 2011 (two days of cumulative rainfall exceeding 365 mm, as shown in Figure 2) triggered approximately 151 shallow landslide events. Most of the landslides transformed into debris-flows, which flowed along the channel in the mountain, reaching cars, roads, and infrastructure. Sixteen casualties were reported, and more than 10 buildings were damaged by the debris, resulting in an economic loss of over 15 million USD. For additional detailed information on the study area, refer to the following papers: Yune et al. [27] and Jeong et al. [28].

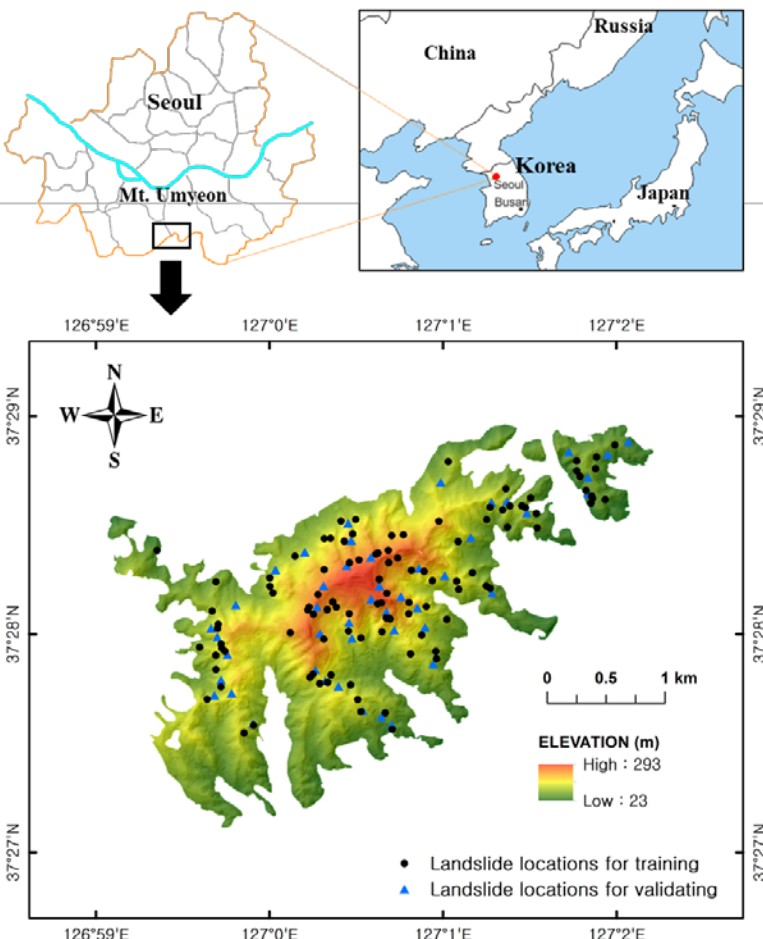

**Figure 1.** Landslide locations in the study area.

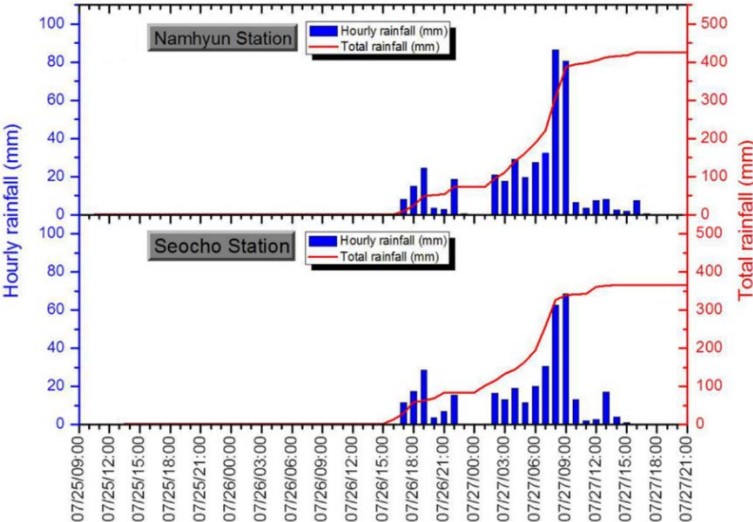

**Figure 2.** Hourly and total rainfall distribution on 25–27 July 2011 at Namhyun and Seocho rainfall station [29].

## 2.2. Landslide Inventory

Collecting an accurate landslide inventory is the most important step in the development of reliable and efficient landslide susceptibility models. In this study, 51-cm-resolution digital orthographic images (Figure 3) provided by the National Geographic Information Institute (NGII) were used to identify locations of landslide initiation. One hundred and fifty-one shallow landslide locations were determined by comparing these images before and after the 2011 landslide events (Figure 3). A landslide inventory map was produced as feature points using ArcMap version 10.6.1 (Figure 1).

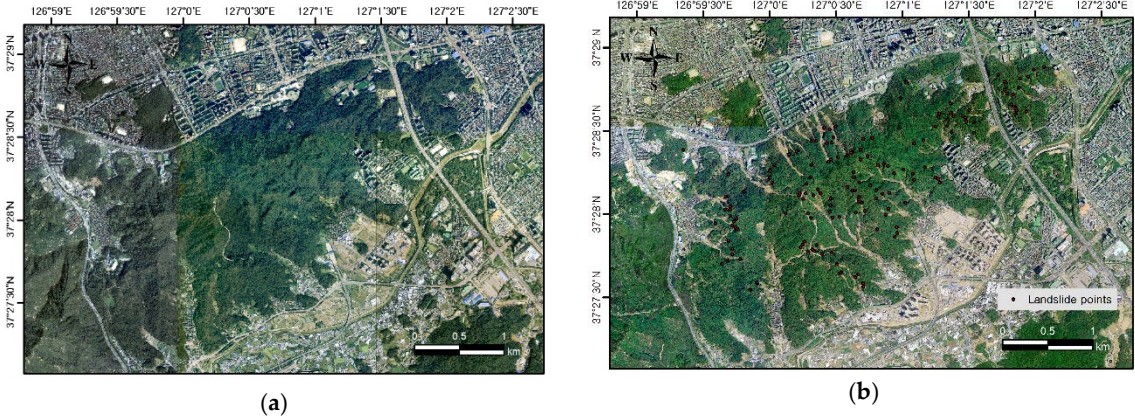

**Figure 3.** Digital orthoimages of the study area: (**a**) before landslides in 2011; (**b**) after landslides in 2011.

## 2.3. Landslide Predisposing Factors

Landslides occur due to complex interactions between various geo-environmental factors. A total of 20 landslide predisposing variables were selected based on abundant literature review and were categorized into four types (morphological, hydrological, geological, and land covers types) [12] as shown in Table 1 and Figure 4. The Digital Elevation Model (DEM), which was provided with a 1:5000 scale by NGII, is fundamental data that is converted into morphological and hydrological variables. Geological and land cover variables were obtained from a 1:25,000 scale forest soil map produced by the Korea Forest Service (KFS). All landslide predisposing candidates were constructed with a 5 × 5 m resolution using ArcMap version 10.6.1 (Esri, Redlands, CA, USA) and consisted of 239,280 grid

cells. Frequency ratio (FR) analysis was performed to evaluate the relationships between landslide occurrence and the predisposing factors, as shown in Table A1.

**Table 1.** Spatial database for the landslide susceptibility map.

| Type | Factor | Source | Scale (Resolution) | Organization |
|---|---|---|---|---|
| | Landslide Inventory | Digital Ortho Images | 51 × 51 cm | NGII |
| Morphological | Elevation | DEM | 1:5,000 | NGII |
| | Slope | | (5 × 5 m) | |
| | Aspect | | | |
| | Curvature | | | |
| | TRI | | | |
| | SRR | | | |
| | SEI | | | |
| Hydrological | TWI | DEM | 1:5,000 | NGII |
| | STI | | (5 × 5 m) | |
| | SPI | | | |
| | Distance from stream | | | |
| Geological | Lithology | Forest Soil Map | 1:25,000 | KFS |
| | Weathering | | (5 × 5 m) | |
| Land Cover | Soil effective depth | Forest Soil Map | 1:25,000 | KFS |
| | Soil type | | (5 × 5 m) | |
| | Soil texture | | | |
| | Soil density | | | |
| | Forest type | | | |
| | Forest density | | | |
| | Distance from road | | | |

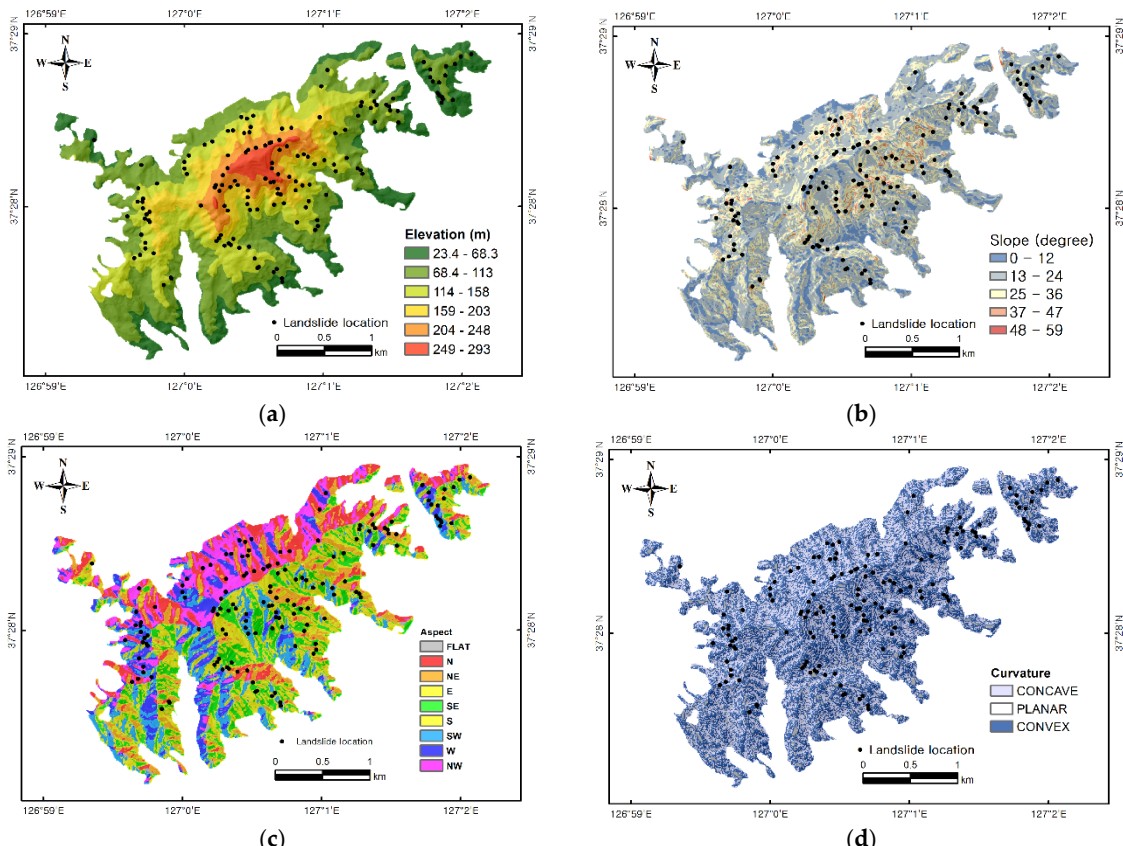

**Figure 4.** *Cont.*

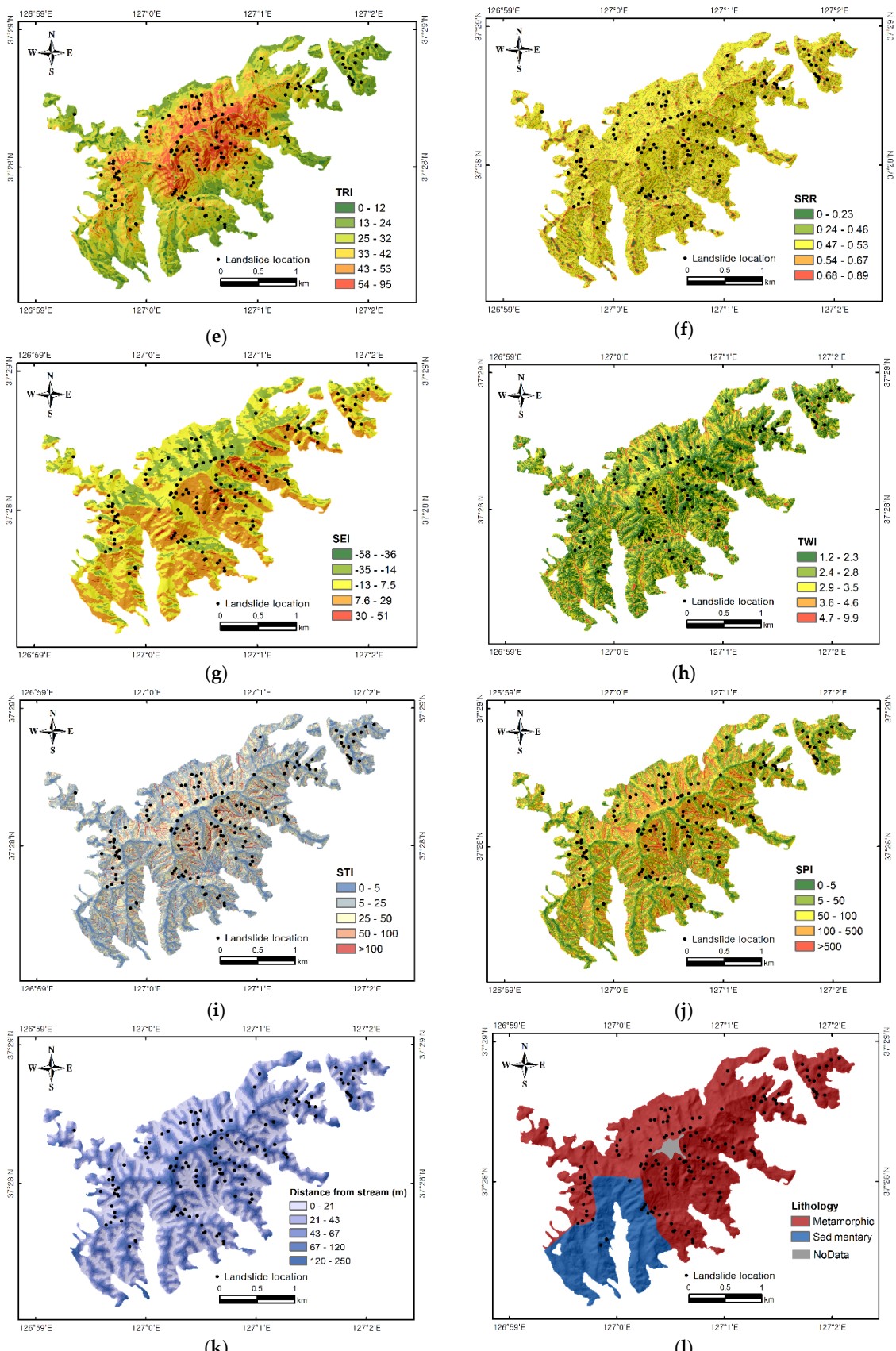

**Figure 4.** *Cont.*

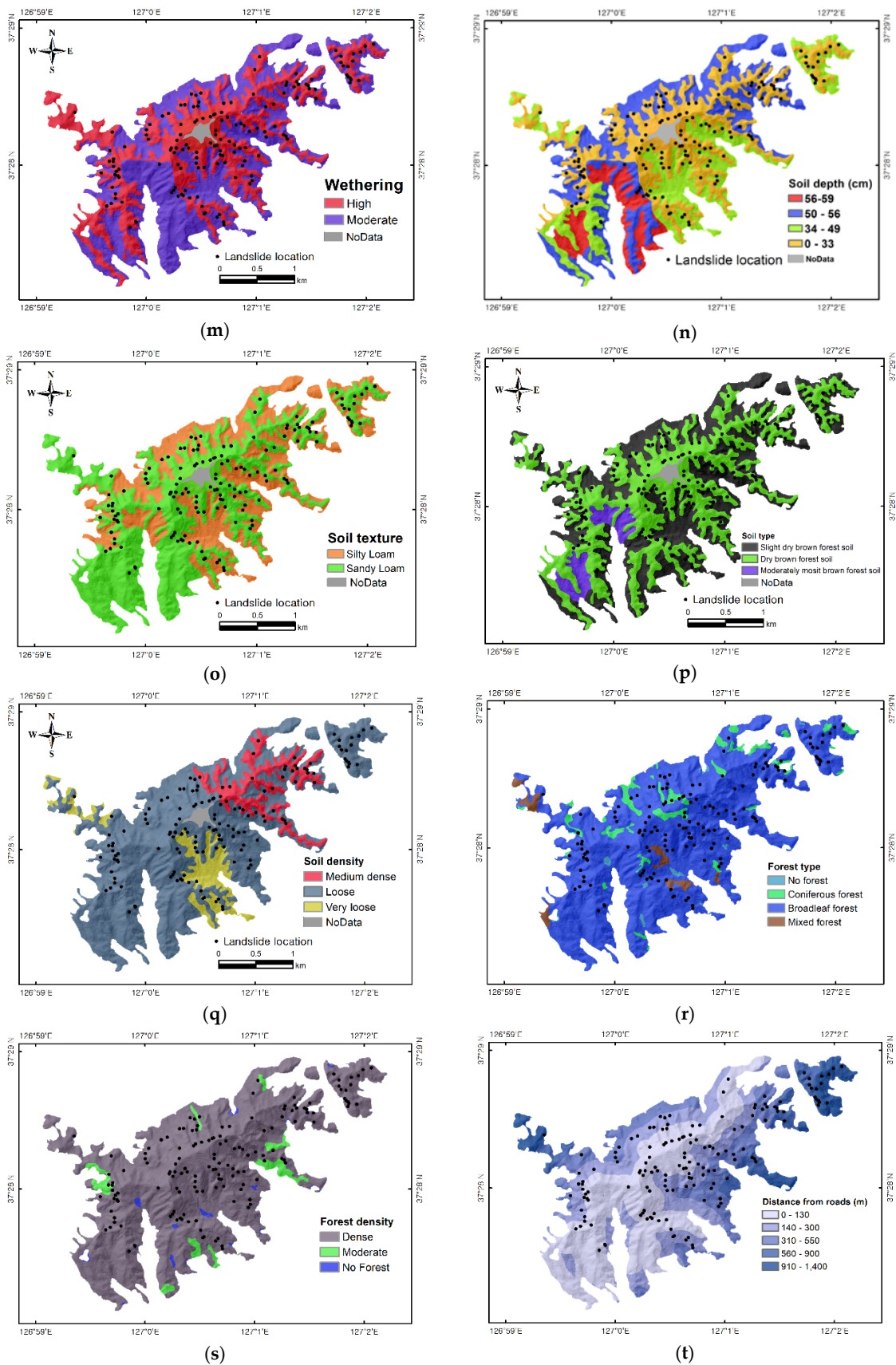

**Figure 4.** Landslide predisposing factors: (**a**) elevation; (**b**) slope; (**c**) aspect; (**d**) curvature; (**e**) topographic ruggedness index (TRI); (**f**) surface relief ratio (SRR); (**g**) site exposure index (SEI); (**h**) topographic wetness index (TWI); (**i**) sediment transport index (STI); (**j**) stream power index (SPI); (**k**) distance from stream; (**l**) lithology; (**m**) weathering; (**n**) effective soil depth; (**o**) soil texture; (**p**) soil type; (**q**) soil density; (**r**) forest type; (**s**) forest density; (**t**) distance from road.

### 2.3.1. Morphological Types

Many studies have shown that the occurrence of landslides is affected by morphological factors, such as elevation, slope, curvature, topographic ruggedness index (TRI), surface relief ratio (SRR), aspect, and site exposure index (SEI). In this study, elevation is in the range of 23.4–293 m. Approximately half of the shallow landslides occurred at the middle elevation of the mountain between 104–203 m, whereas fewer shallow landslides developed at lower elevations. Slope is well known as the most critical cause of landslide occurrence. The slope distribution ranges from 0° to 59°, and the number of landslide events increased with increasing slope angle, with no landslides occurring in the case of slope angles less than 13°. This is due to the shear stress of the soil being directly affected by the slope angle. Curvature is the rate of angular change that indicates the bending degree of a line or surface and affects the deceleration and acceleration of the flowing surface water. The values are continuous data: 0 for planar, negative for concave, and positive for convex. TRI describes the difference in elevation values between a center cell ($c_0$) and others surrounding it [30]. It can be calculated using the following equation:

$$\text{TRI} = \sqrt{\sum (c_i - c_0)^2}, \tag{1}$$

where $c_i$ is the elevation of each neighbor cell to $c_0$. The results of the FR analysis showed that the occurrence of landslides increased as the TRI value increased (Table A1). SRR indicates rugosity considering the maximum, minimum, and mean elevation of each grid [31]. It can be calculated as follows:

$$\text{SRR} = \frac{z_{mean} - z_{\min}}{z_{\max} - z_{\min}}, \tag{2}$$

where $z_{\max}$, $z_{\min}$, and $z_{mean}$ are the maximum, minimum, and mean elevations, respectively. Aspect is the main direction of the slope to which the pixel of the DEM data belongs, expressed in angular units (e.g., 0° in the north direction and 180° in the south direction). The angle is measured from the north to the slope in the clockwise direction. SEI is the rescaling aspect to a north/south axis by multiplying the slope. It is described as the relative degree of sun exposure from coolest to warmest locations according to the following equation:

$$\text{SEI} = slope \cdot \cos\left(\pi \cdot \frac{aspect - 180}{180}\right) \tag{3}$$

Aspect and SEI are important in determining soil water content and factors affecting vegetation in relation to sun exposure [32].

### 2.3.2. Hydrological Types

In this study, we considered topographic wetness index (TWI), sediment transport index (STI), stream power index (SPI), and distance from stream as hydrological factors for evaluating landslide susceptibility. TWI is frequently used to quantify soil moisture, which greatly influences landslide occurrence. This is due to the potential decrease in soil strength caused by increased pore water pressure, which is the main cause of landslide initiation [33]. TWI is given by the equation

$$\text{TWI} = \ln\left(\frac{\alpha}{\tan \beta}\right), \tag{4}$$

where $\alpha$ is the local upslope draining area that indicates the amount of water flowing through a certain point, and $\tan \beta$ is the local slope [34]. STI is a measure of the sedimentation transport capacity that represents the possibility of potential landslides. STI can be calculated by the equation

$$\text{STI} = \left(\frac{A_s}{22.13}\right)^{0.6}\left(\frac{\sin \beta}{0.0896}\right)^{1.3}, \tag{5}$$

where $A_s$ is the specific catchment area, and $\beta$ is the local slope angle in degrees [35]. SPI is an indicator of the erosive power of flowing water and increases with surge flowing caused by large upslope draining areas and steep slopes. SPI can be calculated using the equation

$$\mathrm{SPI} = A_s \cdot \tan \beta, \qquad (6)$$

where $A_s$ is the specific catchment area, and $\beta$ is the local slope angle in degrees [36]. The results of FR analysis showed that the trend of landslide occurrence increased as the values of STI and SPI increased within the range of 0 to 100 (Table A1). Distance from stream was used to assess landslide susceptibility as it may influence rainfall drainage and runoff processes. It was measured according to the Euclidean distance method in ArcGIS 10.6.1, although the results indicated that no landslides occurred more than 120 m away from the stream.

### 2.3.3. Geological Types

Geological features play an important role in landslide susceptibility as such factors can involve a variety of soil and rock properties such as strength, structure, fracture, and composition. In this study, lithology and weathering level were selected as geological features, which are seldom used in the development of landslide susceptibility maps [37]. The lithology of this area consists of metamorphic (75%) and sedimentary (24%) rocks. The weathering level in this area is high (46%) or moderate (53%). The results of FR analysis indicate that landslides occurred predominantly in metamorphic areas and areas with a high level of weathering (Table A1).

### 2.3.4. Land Cover Types

Land cover factors have also been recognized as significant causes that affect slope instability in landslide-prone areas [38,39]. In this study, effective soil depth, soil texture, soil type, soil density, forest type, forest density, and distance from roads were chosen as seven candidate predisposing factors of landslide susceptibility. The effective soil depth ranged from 1 cm to 69 cm, and shallower soil depths induced higher frequencies of landslide occurrence (Table A1). This is due to the fact that, assuming the same permeability, thinner soil layers require a shorter time to become saturated soil layers: then, the saturated soil causes slope instability due to the reduced shear stress. The soil texture of the study area consists of silty loam (38%) and sandy loam (61%), and the soil type is composed of dry brown forest soil (49%), slight dry brown soil (45%), and moderately mist brown forest soil (5%). The results of the FR analysis showed that landslides occurred mainly in sandy loam and dry brown forest soil (Table A1). The soil density of the study area was mainly loose density (79%), but medium dense soil (11%) showed higher FR values (=1.35) compared to loose dense areas (FR=0.97). Forest type and density are important factors as trees can affect slope stability due to root strength, water adsorption, and tree weight [40]. The forest type in this area consists of coniferous (2%), broadleaf (90%), and mixed forest (6%) with mostly dense forest areas. The distance from road was considered a landslide predisposing candidate to evaluate landslide susceptibility as man-made roads in mountains may be a potential cause for slope instability.

## 3. Methodology

A landslide susceptibility analysis was performed through seven main processes as follows: (1) collection of historical landslide data; (2) construction of landslide predisposing factors; (3) preparation of training and validation datasets; (4) application of a filter method to select suitable database subsets; (5) development of landslide susceptibility models; (6) validation and comparison of landslide susceptibility models; (7) selection of the best performing model. Steps (1) and (2) were described earlier in Section 2, and the remaining steps are described below.

### 3.1. Preparation of Training and Validation Datasets

Supervised learning, including the ANN model, requires the preparation and preprocessing of input-target pairs. Targets in landslide susceptibility analysis methods are generally classified into two classes: landslide occurrence (assigned as "1") and non-landslide occurrence (assigned as "0"). A total of 20 landslide predisposing factors were considered as input variables in this study (Figure 4). The continuous variables were rescaled in the range of 0.01 to 0.99 using the min-max normalization formula [22] as follows:

$$z = \frac{x - \min(x)}{\max(x) - \min(x)} (U - L) + L,$$

(7)

where $z$ is the normalized value, $x$ is the original value, and $U$ and $L$ are the upper and lower normalization bounds, respectively. The nominal variables were calculated as frequency ratios (Table A1) and were also normalized using the same method as mentioned earlier.

Preprocessed input-target data for landslide susceptibility modeling should be divided into training and validation datasets. The training dataset is used for model generation, whereas the validation dataset (not the data used for training) is used to validate the developed models and confirm the predictive ability and accuracy of each model. Although there are no exact standards for dividing the two data subsets, this study divided the training and validation subsets according to a 70:30 ratio. Of the total 151 landslide points, 106 and 45 landslide points were randomly split between the training and validation subsets, respectively (Figure 1). The same number and ratio of non-landslide points were also randomly selected from areas safe from landslides. Finally, the values of the 20 landslide predisposing factors were extracted to build the training and validation datasets from the landslide and non-landslide points.

### 3.2. Landslide Predisposing Factor Analysis

The landslide predisposing factor analysis was conducted to select suitable factors for producing landslide susceptibility maps, which is known to be useful in constructing and simplifying machine learning models [41]. Among the various factor selection techniques, we used the filter method, which is an approach to evaluate the relationship between input variables through mathematical and statistical measures. In this study, information gain ratio analysis and multicollinearity analysis were performed among the filter-based factor selection methods.

#### 3.2.1. Information Gain Ratio Analysis

Information gain ratio (IGR) is widely used as a factor selection technique of landslide predisposing factors when quantifying importance based on information theory [13,42,43]. Landslide predisposing factors with high IGR values indicate high predictive power for landslide susceptibility modeling. On the contrary, landslide predisposing factors with low IGR values exhibit low predictive power, which may adversely affect the performance of the susceptibility model. Therefore, it is necessary to select appropriate factors for high performance-based susceptibility model generation.

The basic principle of IGR is as follows. Let a training dataset $S$ be a set consisting of $n$ input variables, and $n(C_i, S)$ is the number of variables in $S$ belonging to the class $C_i$ (landslide, non-landslide). The important quantity (referred to as information or entropy) of $S$ is given as follows:

$$Info(S) = -\sum_{i=1}^{2} \frac{n(C_i, S)}{|S|} \log_2 \frac{n(C_i, S)}{|S|}$$

(8)

The amount of information based on the division of *S* into subsets ($S_1$, $S_2$, ... ,$S_m$) regarding the landslide predisposing factor *L* is calculated as follows:

$$Info(S,L) = \sum_{j=1}^{m} \frac{S_j}{|S|} Info(S) \tag{9}$$

Then, the IGR for landslide predisposing factor *L* is estimated as follows:

$$IGR(S,L) = \frac{Info(S) - Info(S,L)}{SplitInfo(S,L)}, \tag{10}$$

where *SplitInfo* represents the entropy generated by splitting the training data *S* into *m* subsets. *SplitInfo* is defined as

$$SplitInfo(S,L) = -\sum_{j=1}^{m} \frac{|S_j|}{|S|} \log_2 \frac{|S_j|}{|S|} \tag{11}$$

### 3.2.2. Multicollinearity Analysis

Multicollinearity refers to a phenomenon in which certain predisposing factors have a strong correlation with other factors and thus have a negative effect on model accuracy and quality. To resolve this problem, it is necessary to analyze the correlation between predisposing factors through multicollinearity diagnosis when developing statistical or machine learning models. There are several methods of detecting multicollinearity such as the variance inflation factor (VIF) and the tolerance analysis [44], the Farrar–Glauber test [45], the condition number test [46], and the bivariate correlation analysis. In this study, we used Pearson's correlation, VIF, and the tolerance analysis, which are commonly considered for multicollinearity diagnosis in landslide studies [47,48].

Pearson's correlation method was used in this study to confirm the correlation between individual landslide predisposing factors. Pearson's correlation coefficient (*r*) is defined as the covariance of two factors divided by the product of the standard deviations, as follows:

$$r_{XY} = \frac{\sum_{i=1}^{n} (X_i - \overline{X})(Y_i - \overline{Y})}{\sqrt{\sum_{i=1}^{n} (X_i - \overline{X})^2} \sqrt{\sum_{i=1}^{n} (Y_i - \overline{Y})^2}}, \tag{12}$$

where *X* and *Y* are the landslide predisposing factors, and $\overline{X}$ and $\overline{Y}$ are the means of *X* and *Y*, respectively. An *r* value higher than 0.7 indicates a high correlation between *X* and *Y*, whereas an *r* value lower than 0.3 indicates a low correlation.

VIF measures the variation of standard deviation and is increased due to collinearity between the landslide predisposing factors. VIF is calculated as follow:

$$VIF = 1/tolerance = \frac{1}{1 - R^2}, \tag{13}$$

where $R^2$ is the coefficient of determination. The magnitude of multicollinearity can be analyzed through the size of VIF and tolerance. The cutoff values of VIF and the tolerance in this study were 10 and 0.1, respectively.

### 3.3. Landslide Susceptibility Analysis

#### 3.3.1. Artificial Neural Networks

Biological brains store and learn information by sending and receiving signals through synapses that connect neurons to nerve cells. An artificial neural network (ANN) is an algorithm created to mimic how information processing is performed by the human brain, which performs complex

computations by connecting multiple neurons. In this study, the multi-layer perceptron (MLP) model, which is the most widely used ANN model in landslide studies [13,18,22,49], was used to estimate the non-linear relationships between shallow landslides and the predisposing factors. The MLP model generally consists of an input layer, one or more hidden layers, an output layer, and the connection of neurons, as illustrated in Figure 5.

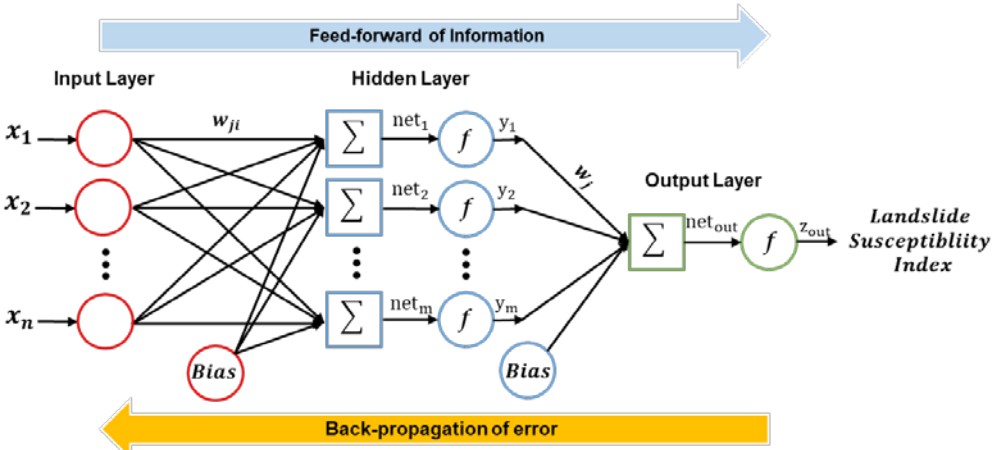

**Figure 5.** Architecture of the multi-layer perceptron (MLP) model.

The input layer of the network, or the first layer, provides information from the outside to the network. The number of neurons in the input layer equals to the number of landslide predisposing factors. The neurons pass the input data on to new neurons in the hidden layer. In the hidden and output layers, the net is calculated as the sum of the products of each weight and bias, and then the output value is calculated by inputting the net value to the activation functions, as described in Equations (14) and (15):

$$y_j = f(net_j) = f\left(\sum_{i=1}^{n} w_{ji} x_i + b_i\right), \tag{14}$$

$$z_{out} = f(net_{out}) = f\left(\sum_{j=1}^{m} w_j y_j + b_j\right), \tag{15}$$

where $x_i$ is the input value, $y_i$ and $z_{out}$ are the output values of the hidden and output layers, respectively, $w_{ji}$ and $w_j$ are the synaptic weights, $b_i$ and $b_j$ are the biases, n and m are the number of neurons in the input and hidden layers, respectively, and $f$ is an activation function, such as linear and non-linear functions. The complexity of the model depends on the number of neurons in the hidden layer, which is determined by the training and testing results of the MLP model.

The learning procedure of MLP is divided into two main processes: i) feed-forward and ii) back-propagation. For the feed-forward phase, the input value is propagated to the output layer and all weights in the network are randomly assigned, resulting in predictive value. In the subsequent back-propagation phase, the weights are updated to minimize the difference between the predicted value and actual value of the network through the gradient descent method. This process is repeated until a mean square error (MSE) value reaches a certain threshold (less than 0.01). In this study, the ANN model was trained using Bayesian regularization [50,51], which is commonly used for error back-propagation algorithms [22,49]. Initial training parameters were set to the default values of MATLAB version R2017a.

### 3.3.2. Activation Functions

The role of the activation function is to add non-linearity to the network by deciding whether a neuron should be activated or not, which makes ANN models capable of learning and performing

complex phenomena. Therefore, selecting the appropriate activation function is an essential task involved in predicting complex phenomena. As shown in Figure 5, in the MLP model, the activation function is used in the hidden and output layers. In this study, we used one hidden layer structure for a simple comparison between activation functions, so that a total of two activation functions are applied to each of the hidden and output layers.

There are several types of activation functions, each of which has its own various advantages and limitations. So far, no activation function stands out as having the best performance overall other functions, and thus, activation functions need to be selected based on how well it matches the characteristics of the implemented model. Activation functions can be categorized into two types: i) linear and ii) non-linear. A linear activation function is a function in which the output is the product of an input value multiplied by a certain constant value, described in Equation (16),

$$f(x) = cx, \tag{16}$$

where $x$ is the input value and $c$ is a constant.

In contrast, there are several types of non-linear activation functions. This study utilizes functions that are commonly used, as follows: i) the symmetric hard limit (Hard-lims) function, ii) the symmetric saturating linear (Sat-lins) function, iii) the radial basis (Rad-bas) function, iv) the logistic sigmoid (Log-sig) function, v) the rectified linear unit (ReLU) function, and vi) the hyperbolic tangent sigmoid (Tan-sig) activation function.

The Hard-lims function is a binary function that sends a signal of 1 for positive and –1 for negative values, which can be expressed as follows:

$$f(x) = \begin{cases} 1, x \geq 0 \\ -1, x < 0 \end{cases} \tag{17}$$

The Sat-lins function has identical outputs for inputs in the range of −1 to 1, an output value of 1 if the input is greater than 1, and an output value of −1 if the input is less than −1, which can be expressed as follows:

$$f(x) = \begin{cases} 1, x > 1 \\ x, -1 \leq x \leq 1 \\ -1, x < 0 \end{cases} \tag{18}$$

The Rad-bas function is expressed in Equation (19). A Gaussian function is generally used for the Radbas function. It has an output value that increases or decreases monotonically with distance from the center point.

$$f(x) = \exp\left(-\frac{(x-c)^2}{r^2}\right), \tag{19}$$

where $c$ is the center and $r$ is the radius. The ReLU function has an identity for all positive values, and zero for all negative values, which can be expressed as follows:

$$f(x) = \begin{cases} x, x \geq 0 \\ 0, x < 0 \end{cases} \tag{20}$$

The Log-sigmoid function is given Equation (21), and outputs values ranging between 0 and 1. Larger inputs converge to 1, and smaller inputs converge to 0, hence the output is not zero-centered.

$$f(x) = \frac{1}{1 + e^{-x}} \tag{21}$$

The tan-sigmoid function is given by Equation (22). It has an output range from −1 to 1. The shape of the tan-sigmoid function is loosely similar to log-sigmoid as an S-shape, but the tan-sigmoid function is centered on zero.

$$f(x) = \frac{e^x - e^{-x}}{e^x + e^{-x}} \tag{22}$$

Users of MLP models must decide on which activation functions to use for the hidden and output layers. In this study, the activation functions of the hidden layer were the six non-linear types of functions (Hard-lims, Sat-lins, Rad-bas, ReLU, Log-sig, and Tan-sig). For the output layer, the Log-sig activation function, which is frequently used for binary classification, was used to express the result between 0 and 1. In this case, the closer the result is to 1, the higher the probability of a landslide. In contrast, the closer the result is to 0, the lower the probability of a landslide. In the case of the linear activation function, as it is mainly used in the output layer to develop regression models, it was excluded from this study.

### 3.4. Assessment of Model Performance

#### 3.4.1. Statistical Evaluation Measures

The performance of landslide susceptibility models can be evaluated using various statistical measures. In this study, we used sensitivity, specificity, accuracy, positive predictive value (PPV), negative predictive value (NPV), and Kappa index to evaluate the performance of the models. Sensitivity measures the proportion of landslide pixels that are correctly identified as landslide occurrences. Specificity measures the proportion of non-landslide pixels that are correctly identified as non-landslide occurrences. Accuracy measures the proportion of landslide and non-landslide pixels that are correctly identified. Positive predictive value is the probability that predicted landslide pixels have actual landslide occurrences. Negative predictive value is the probability that predictive non-landslide pixels have actual non-landslide occurrences. The Kappa index, which is generally regarded as a reliability measure, is used to measure the quality of classification models [52]. In other words, it can be used to evaluate how effectively a landslide susceptibility model classifies landslide pixels [53]. According to Landis and Koch [54], the degree to which the observed and predicted values are in agreement in terms of the Kappa index are summarized as follows: 0.81–1.00 indicates near-perfect agreement, 0.61–0.80 indicates substantial agreement, 0.41–0.60 indicates moderate agreement, 0.21–0.40 indicates fair agreement, 0.00–0.20 indicates slight agreement, and values lower than 0 indicate poor agreement. These statistical measures can be calculated from a confusion matrix and are expressed as follows:

$$\text{Sensitivity} = \frac{TP}{TP + FN}, \tag{23}$$

$$\text{Specificity} = \frac{TN}{TN + FP}, \tag{24}$$

$$\text{Accuracy} = \frac{TP + TN}{TP + NT + FP + FN} \tag{25}$$

$$PPV = \frac{TP}{FP + TP} \tag{26}$$

$$NPV = \frac{TN}{FN + TN} \tag{27}$$

$$\text{Kappa index} = \frac{p_{observed} - p_{expected}}{1 - p_{expected}} \tag{28}$$

where *TP* (true positive) and *TN* (true negative) are the numbers of correctly identified pixels, whereas *FP* (false positive) and *FN* (false negative) are the numbers of pixels that were incorrectly identified.

$p_{observed}$ is identical to accuracy, and $p_{expected}$, which is expressed by Equation (29), is the expected proportion of landslide and non-landslide pixels that are in agreement.

$$p_{expected} = \left(\frac{TP + FP}{All}\right)\left(\frac{TP + FN}{All}\right) + \left(\frac{FN + TN}{All}\right)\left(\frac{FP + TN}{All}\right), \tag{29}$$

where *All* is the summation of the number of correctly identified pixels and incorrectly identified pixels, which is calculated as *TP+TN+FN+FP*.

### 3.4.2. Receiver Operating Characteristic Curve

The receiver operating characteristic (ROC) curve is widely used to confirm the performance of landslide susceptibility models. The curve is a plot of the true positive rate (*=Sensitivity*) against the false positive rate (*=1–Specificity*) with various cut-off settings. The area under the ROC curve (AUC) can be used for quantitative comparisons of model performance. The closer the AUC value is to 1, the higher the capability of the model to distinguish between landslides and non-landslides, whereas when AUC is 0.5, the model is said to have no capability in separating landslides and non-landslides.

### 3.4.3. Non-Parametric Statistical Test

A non-parametric test is a statistical method of confirming statistical significance based on given data without assuming the probability distribution of the parameters. Such tests were used to confirm statistically significant differences among the developed landslide susceptibility models. In this study, the Friedman test was considered to compare the performance of the landslide susceptibility models. This non-parametric test is based on the null hypothesis, in which no significant differences exist between the performances of the landslide susceptibility models. If the significant probability (p-value), which is the probability of obtaining test results with significant differences, is greater than a certain value (5%, 0.05), the null hypothesis is accepted. Conversely, if the p-value is lower than 0.05, then the null hypothesis is rejected.

## 4. Results

### *4.1. Landslide Predisposing Factor Analysis*

#### 4.1.1. Predictive Ability Analysis

In this study, IGR analysis of the landslide predisposing factors, which is the first factor-selecting step, was conducted to check the predictive ability of each factor. The results show that slope is the most significant landslide predisposing factor with an average merit value of 0.38, followed by STI (0.35), TRI (0.34), SPI (0.30), SRR (0.30), aspect (0.25), TWI (0.23), soil depth (0.23), soil type (0.20), curvature (0.20), lithology (0.15), SEI (0.11), and elevation (0.10), as shown in Figure 6. The remaining seven predisposing factors (forest density, forest type, soil texture, soil density, weathering, distance from stream, and distance from road) were analyzed to be infinitesimal landslide predisposing factors with an average merit value of 0. Therefore, in this study, 13 landslide predisposing factors with an average merit value greater than 0 were selected among the 20 factors, and the remaining seven factors were eliminated.

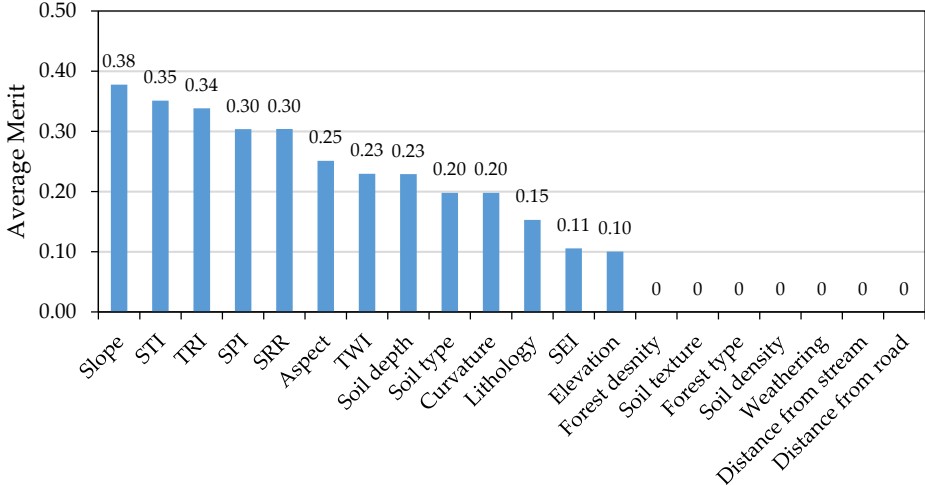

**Figure 6.** Average merit of the landslide predisposing factors.

### 4.1.2. Multicollinearity Diagnostic Analysis

Selected 13 landslide predisposing factors were performed for multicollinearity using Pearson's correlation, VIF and tolerance analysis. In the first step, Pearson's correlation analysis was conducted between pairs of landslide predisposing factors. The results show that Elevation with TRI (0.76), Slope with TRI (0.80), and STI with SPI (0.83) pairs have a high correlation value, as shown in Table 2. If these values exceed 0.7, it can be suspected that there is multicollinearity. The TRI represents the ruggedness of the terrain through the altitude difference of neighboring cells, so it can be explained that it correlates with the elevation and the slope. STI and SPI are also calculated based on the upslope contributing area and the slope value, which explains the high correlation between them. In these cases, TRI and SPI variables with relatively lower average merit values were removed from the landslide predisposing factor candidates.

**Table 2.** Pearson's correlation between pairs of landslide predisposing factors.

|         | EL [1] | SL [2] | AS [3] | CU [4] | TRI   | SRR   | SEI  | TWI   | STI   | SPI   | LI [5] | SD [6] |
|---------|--------|--------|--------|--------|-------|-------|------|-------|-------|-------|--------|--------|
| EL [1]  | 1      |        |        |        |       |       |      |       |       |       |        |        |
| SL [2]  | 0.31   | 1      |        |        |       |       |      |       |       |       |        |        |
| AS [3]  | 0.12   | 0.10   | 1      |        |       |       |      |       |       |       |        |        |
| CU [4]  | 0.16   | 0.21   | 0.16   | 1      |       |       |      |       |       |       |        |        |
| TRI     | *0.76* | *0.80* | 0.13   | 0.10   | 1     |       |      |       |       |       |        |        |
| SRR     | 0.06   | 0.07   | 0.07   | 0.18   | 0.11  | 1     |      |       |       |       |        |        |
| SEI     | 0.04   | 0.06   | 0.06   | 0.33   | 0.00  | 0.01  | 1    |       |       |       |        |        |
| TWI     | −0.07  | −0.42  | 0.01   | 0.06   | −0.38 | −0.24 | 0.09 | 1     |       |       |        |        |
| STI     | 0.11   | 0.23   | 0.07   | −0.07  | 0.22  | 0.00  | 0.03 | 0.47  | 1     |       |        |        |
| SPI     | −0.01  | −0.05  | 0.02   | −0.06  | −0.03 | 0.00  | 0.02 | 0.40  | *0.83*| 1     |        |        |
| LI [5]  | −0.10  | −0.08  | −0.03  | 0.05   | −0.12 | −0.02 | 0.21 | 0.02  | −0.07 | −0.03 | 1      |        |
| SD [6]  | −0.27  | −0.11  | 0.02   | 0.06   | −0.23 | −0.05 | 0.18 | 0.24  | 0.12  | 0.11  | 0.64   | 1      |
| ST [7]  | 0.34   | 0.11   | 0.3    | 0.15   | 0.25  | 0.02  | 0.09 | −0.15 | −0.12 | −0.09 | 0.29   | −0.22  |

[1] Elevation, [2] Slope, [3] Aspect, [4] Curvature, [5] Lithology, [6] Soil depth, and [7] Soil type.

The next step for selecting landslide predisposing factors was to analyze VIF and tolerance. The result shows that the highest value of VIF is 5.426 and the lowest value of tolerance 0.184, as shown in Table 3. These values satisfied critical thresholds, which are VIF > 10 or tolerance < 0.1, which represent no multicollinearity among the 11 landslide predisposing factors. Finally, the best 11 landslide predisposing factors were selected through factor selection based on IGR and multicollinearity analysis.

**Table 3.** Multicollinearity analysis for the landslide predisposing factors.

| Number | Landslide Predisposing Factor | VIF | Tolerance |
|--------|-------------------------------|-------|-----------|
| 1 | Elevation | 1.394 | 0.717 |
| 2 | Slope | 4.570 | 0.219 |
| 3 | Aspect | 1.256 | 0.796 |
| 4 | Curvature | 1.204 | 0.830 |
| 5 | SRR | 2.369 | 0.422 |
| 6 | SEI | 1.071 | 0.933 |
| 7 | TWI | 5.426 | 0.184 |
| 8 | STI | 2.396 | 0.417 |
| 9 | Lithology | 3.763 | 0.266 |
| 10 | Soil depth | 4.462 | 0.224 |
| 11 | Soil type | 2.878 | 0.347 |

*4.2. Landslide Susceptibility Modeling, Validation, and Comparison*

4.2.1. Selecting the Best Number of the Neurons in the Hidden Layer

Using the 11 selected predisposing factors, MLP models consisting of Hard-lims, Sat-lins, Rad-bas, Log-sig, and Tan-sig activation functions for the hidden layer with the Log-sig activation function for the output layer were produced using the training dataset. To determine the optimal number of neurons in the hidden layer, performance evaluation was carried out using classification accuracy with the training and validation datasets. In addition, the Kappa index was determined with the validation dataset. The results of the Hard-lims model showed that as the number of neurons increased, the overall classification accuracy and Kappa index increased, but converged or decreased to a certain value beyond a certain number of neurons. In the case of the Sat-lins and ReLU models, the accuracy and Kappa index tended to rise and fall slightly without significant change as the number of neurons increased. The results of the Rad-bas, Log-sig, and Tan-sig models showed that, as the number of neurons increased, the accuracy with the training dataset increased. On the other hand, the accuracy and Kappa index with the validation dataset decreased or fluctuated beyond a certain number of neurons. The results for the optimal number of neurons in the hidden layer (Figure 7) were as follows: eight neurons for the Hard-lims model (Figure 7a), six neurons for the Sat-lins model (Figure 7b), three neurons for the Rad-bas model (Figure 7c), six neurons for the ReLU model (Figure 7d), four neurons for the Log-sig model (Figure 7e), and five neurons for the Tan-sig model (Figure 7f).

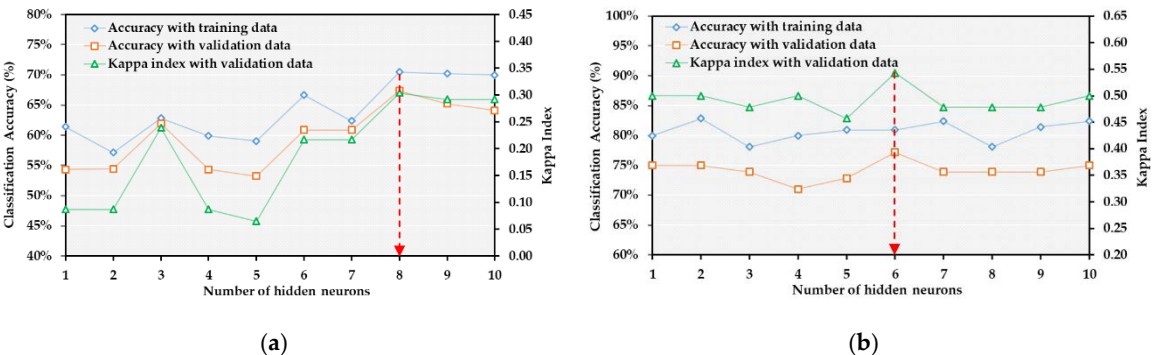

(**a**)                    (**b**)

**Figure 7.** *Cont.*

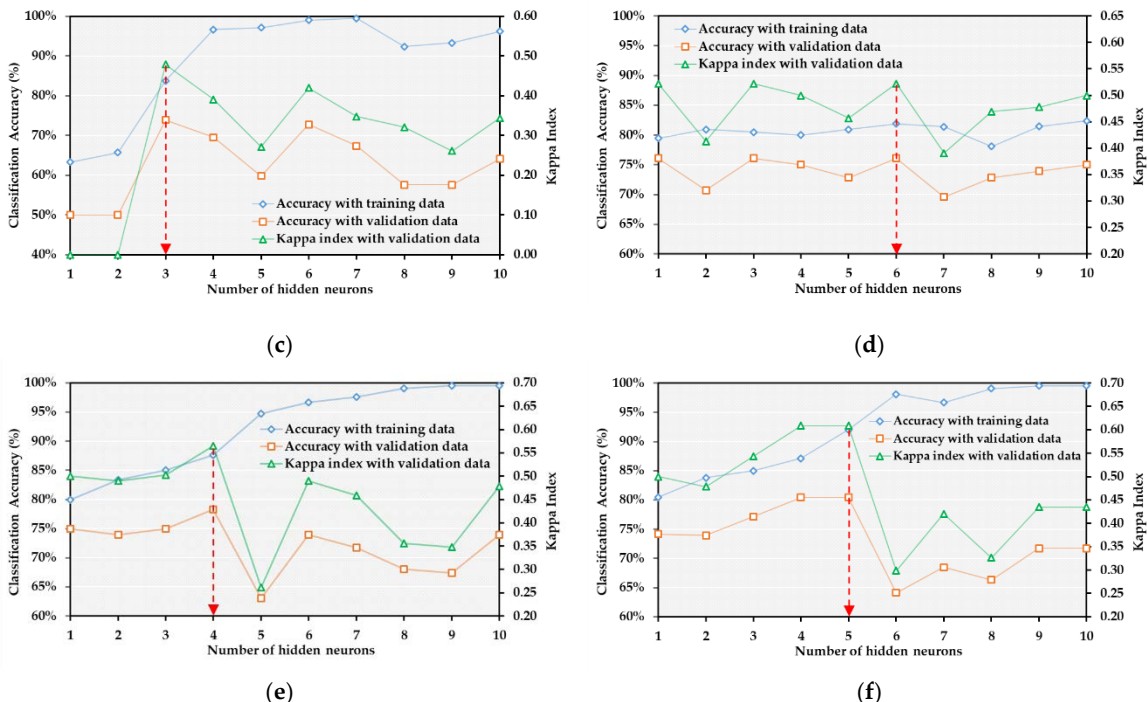

**Figure 7.** Performance of MLP models according to the activation function and the number of neurons: (**a**) Hard-lims, (**b**) Sat-lins, (**c**) Rad-bas, (**d**) ReLU, (**e**) Log-sig, and (**f**) Tan-sig functions.

### 4.2.2. Evaluation of the Model Performance

Performance evaluation of the landslide susceptibility models was performed based on the training data and the optimal number of neurons determined in the previous section. Based on the results of the model evaluation (Table 4), the Tan-sig model exhibited the highest results for overall statistical evaluation indices, whereas the Hard-lims model exhibited the lowest. The Tan-sig model had a sensitivity of 92.38%, which indicates that 92.38% of the landslide pixels were correctly identified as landslide occurrences, followed by the Log-sig model (90.48%), the ReLU model (89.52%), the Rad-bas model (88.57%), the Sat-lins model (86.67%), and the Hard-lims model (68.57%). In addition, the Tangent-sig model exhibited the highest specificity (87.62%), indicating that 87.62% of the non-landslide pixels were correctly identified as non-landslide occurrences, followed by the Log-sig model (84.76%), the Rad-bas model (79.05%), the ReLU model (74.29%), the Hard-lims model (72.38%), and Sat-lins model (68.57%). The highest accuracy value was 90.00% with the Tan-sig model, which indicates that 90.00% of the landslide and non-landslide pixels were correctly identified.

**Table 4.** Results of model performance evaluation.

| Measures | Hard-lims | Sat-lins | Rad-bas | ReLU | Log-sig | Tan-sig |
|---|---|---|---|---|---|---|
| True positive | 72 | 91 | 93 | 94 | 95 | 97 |
| True negative | 76 | 72 | 83 | 78 | 89 | 92 |
| False positive | 29 | 33 | 22 | 27 | 16 | 13 |
| False negative | 33 | 14 | 12 | 11 | 10 | 8 |
| Sensitivity (%) | 68.57 | 86.67 | 88.57 | 89.52 | 90.48 | 92.38 |
| Specificity (%) | 72.38 | 68.57 | 79.05 | 74.29 | 84.76 | 87.62 |
| Accuracy (%) | 70.48 | 77.62 | 83.81 | 81.90 | 87.62 | 90.00 |
| PPV (%) | 71.29 | 73.39 | 80.87 | 77.69 | 85.59 | 88.18 |
| NPV (%) | 69.72 | 83.72 | 87.37 | 87.64 | 89.90 | 92.00 |
| Kappa index | 0.410 | 0.552 | 0.676 | 0.638 | 0.752 | 0.800 |
| AUC | 0.837 | 0.863 | 0.936 | 0.930 | 0.964 | 0.968 |

The highest positive predictive value was 88.18% with the Tan-sig model, which indicates an 88.18% chance of predictive landslide pixels undergoing actual landslide occurrences. This result is followed by a value of 85.59% with the Log-sig model, 80.87% with the Rad-bas model, 77.69% with the ReLU model, 73.39% with the Sat-lins model, and 71.29% with the Hard-lims model. The model with the highest negative predictive value is the Tan-sig with a 92.00% chance of predictive landslide pixels having actual non-landslide occurrences, followed by the Log-sig model with 89.90%, the ReLU model with 87.64%, the Rad-bas model with 87.37%, the Sat-lins model with 83.72% and the Hard-lims model with 69.72%.

The Kappa index values ranged from 0.410 to 0.800 for the six models, indicating that the strength of agreement between the observed and the predicted values of the model was moderate for the Hard-lims model (0.410) and the Sat-lins model (0.552), and substantial for the ReLU model (0.638), the Rad-bas model (0.676), the Log-sig model (0.752), and the Tan-sig model (0.800). All six models had high AUC values, which represents the capability of distinguishing between landslide and non-landslide occurrences; among them, the Tan-sig model had the highest AUC value of 0.968.

4.2.3. Validation of the Model Performance

The results of landslide susceptibility model performance were validated using the statistical evaluation measures based on the validation dataset, as shown in Table 5. The results showed that the Tan-sig model had the highest performance in terms of the overall evaluation indices. The Tan-sig model exhibited the highest values of sensitivity (82.61%), specificity (78.26%), and accuracy (80.43%), which means 82.61% of the landslide pixels were correctly identified as landslide occurrences, 78.26% of the non-landslide pixels were correctly identified as non-landslide occurrences, and 80.43% of the landslide and non-landslide pixels were correctly identified. In addition, the Tan-sig model had the highest positive predictive value and negative predictive value of 79.19% and 81.82%, respectively, indicating a 79.19% chance of predictive landslide pixels having actual landslide occurrences and an 81.82% chance of predictive landslide pixels having actual non-landslide occurrences. The Hard-lims model had the lowest values of all statistical measures due to the relatively low values of true positive and true negative and relatively high values of a false positive and false negative.

**Table 5.** Results of model performance validation.

| Measures | Hard-lims | Sat-lins | Rad-bas | ReLU | Log-sig | Tan-sig |
|---|---|---|---|---|---|---|
| True positive | 32 | 37 | 37 | 36 | 37 | 38 |
| True negative | 28 | 29 | 31 | 34 | 35 | 36 |
| False positive | 18 | 17 | 15 | 12 | 11 | 10 |
| False negative | 14 | 9 | 9 | 10 | 9 | 8 |
| Sensitivity (%) | 69.57 | 80.43 | 80.43 | 78.26 | 80.43 | 82.61 |
| Specificity (%) | 60.87 | 63.04 | 67.39 | 73.91 | 76.09 | 78.26 |
| Accuracy (%) | 65.21 | 71.74 | 73.91 | 76.09 | 78.26 | 80.43 |
| PPV (%) | 64.00 | 68.52 | 71.15 | 75.00 | 77.08 | 79.17 |
| NPV (%) | 30.55 | 76.32 | 77.50 | 77.27 | 79.55 | 81.82 |
| Kappa index | 0.304 | 0.435 | 0.478 | 0.522 | 0.565 | 0.609 |
| AUC | 0.781 | 0.815 | 0.821 | 0.843 | 0.877 | 0.879 |

The Kappa index values ranged from 0.304 to 0.609 for the six models, indicating that the strength of agreement between the observed and predicted values of the models was fair for the Hard-lims model (0.304), moderate for the Sat-lins model (0.478), the Rad-bas model (0.478), the ReLU model (0.522), and the Log-sig model (0.565), and substantial for the Tan-sig model (0.609). The highest value of AUC was 0.879 with the Tan-sig model, which indicates that the model was the most capable of distinguishing between landslide and non-landslide occurrences, followed by the Log-sig model (0.877), the ReLU model (0.843), the Rad-bas model (0.821), the Sat-lins model (0.815), and the Hard-lims

model (0.781). The statistical indices evaluated using the training dataset were higher than the indices evaluated using the validation dataset.

### 4.2.4. Comparison of the Model Performance

The non-parametric Friedman test with a p-value threshold of 5% was performed to compare the performances of the landslide susceptibility models. The results indicated that the p-values were lower than 0.05; therefore, the null hypothesis is rejected, which indicates the existence of statistically significant differences between the performances of the landslide susceptibility models. The results of the six landslide susceptibility models from the Friedman test are shown in Table 6.

**Table 6.** Comparison of the six landslide susceptibility models using the Friedman test.

| Ranking | | | | | | p-value $(\alpha = 0.05)$ | $\chi^2$ (Chi-square) |
|---|---|---|---|---|---|---|---|
| Hard-lims | Sat-lins | Rad-bas | ReLU | Log-sig | Tan-sig | | |
| 3.61 | 3.73 | 3.55 | 3.42 | 3.06 | 3.63 | $1.78 \times 10^{-4}$ | 24.445 |

### 4.2.5. Production of Landslide Susceptibility Maps

All six models were used to create landslide susceptibility maps for the study area. Landslide susceptibility indices (LSI) were divided into five levels as follows: Very High, High, Moderate, Low, and Very Low. The threshold for differentiating the occurrence of landslides and non-landslides was set as an LSI value of 0.5. The intervals of each class were then set as follows: Very High (LSI $\geq$ 0.95), High (0.95 > LSI $\geq$ 0.85), Moderate (0.85 > LSI $\geq$ 0.70), Low (0.79 > LSI $\geq$ 0.50), or Very Low (LSI < 0.50).

Landslide susceptibility maps were produced based on various activation functions (Figure 8). Table 7 shows the calculated percentages of historical landslides according to susceptibility class. The results showed that the Tan-sig model had the following percentages: 35.8% for Very High, 22.5% for High, 31.1% for Moderate, 4.6% for Low, and 6% for Very Low. In addition, the Tan-sig model showed that 89.4% of all historical landslides ranged from the Very High to Moderate classes.

**Table 7.** Percentages of historical landslides according to susceptibility class (unit: %).

| | Hard-lims | Sat-lins | Rad-bas | ReLU | Log-sig | Tan-sig |
|---|---|---|---|---|---|---|
| Very High | 0 | 0 | 16.6 | 5.3 | 27.2 | 35.8 |
| High | 0 | 0 | 17.2 | 20.5 | 21.9 | 22.5 |
| Moderate | 36.4 | 50.3 | 35.1 | 37.8 | 26.2 | 31.1 |
| Low | 32.5 | 37.1 | 17.2 | 22.5 | 15.2 | 4.6 |
| Very Low | 31.1 | 12.6 | 13.9 | 13.9 | 12.6 | 6.0 |
| Total | 100 | 100 | 100 | 100 | 100 | 100 |

Landslide density [55], which is defined as a ratio between the percentage of historical landslide pixels ($P_L$) and the percentage of all areas ($P_{all}$) on the map for a given susceptibility class, is calculated by the following equation:

$$\text{Landslide Density} = \frac{P_L}{P_{all}} \tag{30}$$

The highest value of landslide density was 4.38 for the Very High susceptible class of the Tan-sig model. All landslide density results are summarized in Table 8.

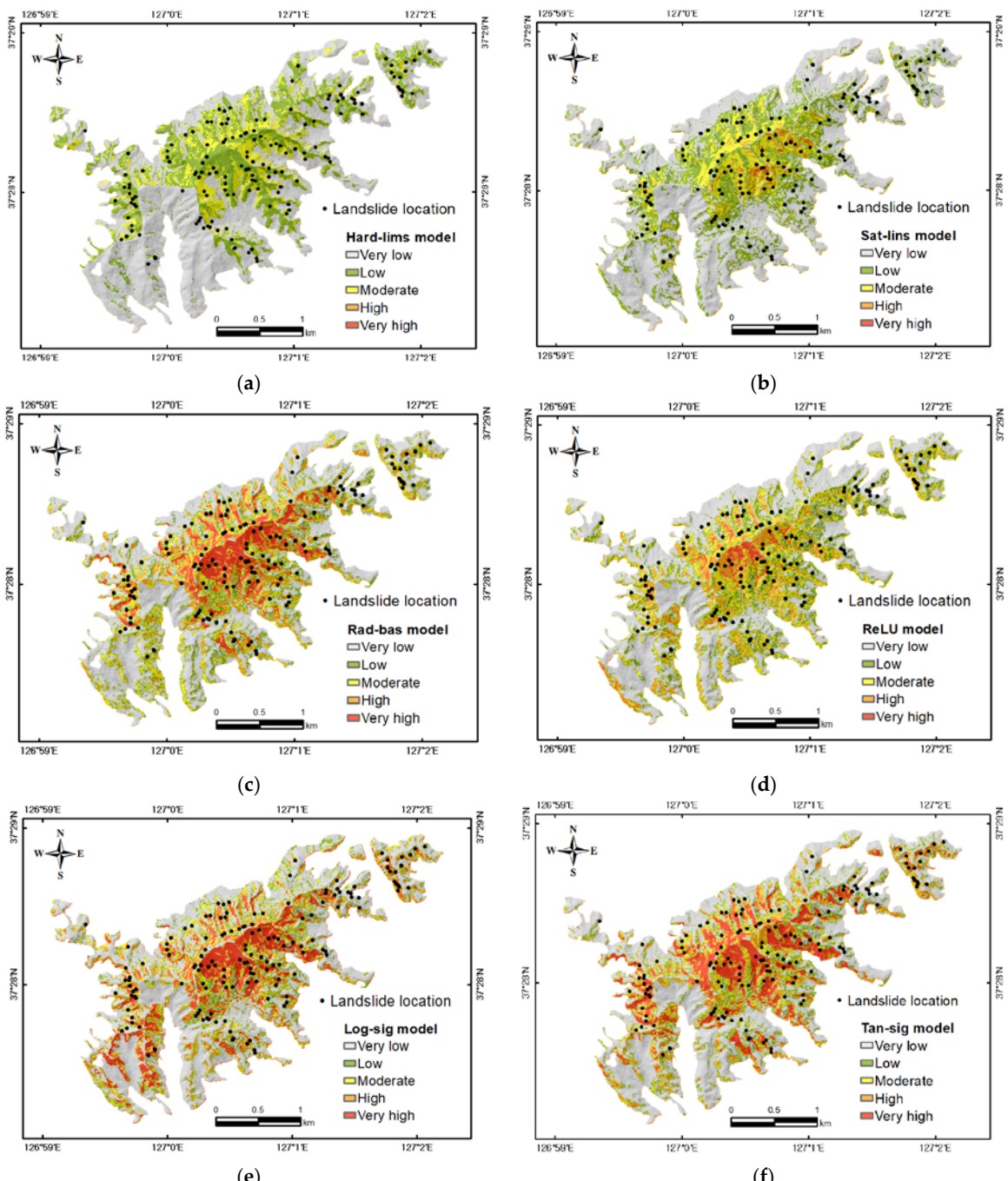

**Figure 8.** Landslide susceptibility maps produced using the (**a**) Hard-lims, (**b**) Sat-lins, (**c**) Rad-bas, (**d**) ReLU, (**e**) Log-sig, and (**f**) Tan-sig functions.

**Table 8.** Landslide density according to susceptibility class.

| Classes | Hard-lims | Sat-lins | Rad-bas | ReLU | Log-sig | Tan-sig |
|---|---|---|---|---|---|---|
| Very High | - | 0.00 | 2.76 | 4.27 | 3.19 | 4.38 |
| High | - | 0.00 | 1.24 | 2.08 | 1.46 | 1.43 |
| Moderate | 2.86 | 3.61 | 1.53 | 1.65 | 1.20 | 1.40 |
| Low | 0.90 | 1.06 | 0.54 | 0.64 | 0.53 | 0.16 |
| Very Low | 0.61 | 0.26 | 0.55 | 0.45 | 0.49 | 0.24 |

## 5. Discussion

Landslide susceptibility mapping is an essential task in the determination of landslide-prone areas and is well recognized as an important step in the prevention and mitigation of landslide hazards. Many researchers have utilized ANN models to develop landslide susceptibility models [15–22]. Despite such attempts, there is still a multitude of considerations involved in determining the optimal structure of a high performance-based ANN model, such as landslide predisposing factor selection, the number of neurons in the hidden layer, and the activation function.

The first important step in developing a landslide susceptibility map involves building a reliable database of input–output pairs as it can control the performance of the susceptibility model. In this study, a landslide inventory was constructed in the form of a feature point with 5 m resolution at the center of the source area. The inventory is able to represent the overall morphological, hydrological, geological, and land cover characteristics of the study area as the landslides that occurred in the study area were shallow and translational or slightly rotational types. For other types of landslides, such as deep failures, it would be more suitable to construct an inventory in the form of feature polygons.

A total of 20 landslide predisposing factors (elevation, slope, aspect, curvature, TRI, SRR, SEI, soil density, forest type, forest density, and distance from road) were established through an abundant literature review of existing landslide susceptibility studies. The established predisposing factors were normalized to a comparable range of 0.01–0.99 for further data analysis and ANN modeling. This process can guarantee stable convergence of weight and biases in ANN modeling [56]. Future studies are recommended to use geotechnical databases such as internal friction angle, cohesion, and permeability coefficient, although such databases may require significant amounts of money and time to build. As such factors directly influence slope stability, reliable research results may be obtained.

Factor selection for assessing landslide susceptibility is an essential task that influences the quality of ANN models, as not all factors affect landslide occurrence. In this study, information gain ratio (IGR), Pearson correlation, VIF, and tolerance analyses were subsequently performed to check the predictive power of each predisposing factor and conduct multicollinearity diagnosis. Although there is no universal agreement regarding factor selection methods, a high-performance ANN model was successfully developed through the method applied in this study.

In the IGR analysis phase, slope showed the highest value of average merit among the predisposing factors, which is judged to be due to its significant contribution to the factor of safety. In contrast, six factors (forest density, soil texture, forest type, soil density, weathering, distance from stream, and distance from road) were determined to possess no predictive ability and were excluded from this study. Nonetheless, the six excluded factors should be further studied in other regions, as these factors may possess predictive power if additional databases are accumulated from different regions. In the multicollinearity diagnosis phase, slope and elevation showed high correlation with TRI as well as STI and SPI. Although high correlation does not necessarily indicate multicollinearity, the calculation formulas of TRI, STI, and SPI indicate high correlation between each variable. Thus, data regarding TRI and SPI were eliminated at low-predictive ability orders. Finally, VIF and tolerance analyses determined that there was no multicollinearity between the 11 selected factors.

The 11 selected predisposing factors were randomly split into a 70:30 ratio for training and validation. Although there are no specific guidelines for dividing datasets, this process may prevent the overfitting or underfitting problem and enable reliable model verifications compared to models that do not divide datasets. The importance of dividing datasets for training and validation was also mentioned and discussed by Chung and Fabbri [57], Tien Bui et al. [22], and several other researchers.

In this study, six models, each with a different non-linear activation function in the hidden layer, were evaluated and validated using the Kappa index, AUC, and five statistical measures. As a result, the best performing MLP model was the model that used the hyperbolic tangent sigmoid (Tan-sig) function with five neurons in the hidden layer (Figure 7, Tables 4 and 5). The models developed with the six activation functions were identified as comparable models by the non-parametric Friedman test, which showed the models as having significant differences with each other (Table 6). Finally, the

Tan-sig model showed that 89.4% of all historical landslides ranged from the Very High to Moderate classes and produced a landslide density result of 4.38 for the Very High susceptible class, which is the highest value among the six models (Tables 7 and 8). In other words, a Tan-sig function in the hidden layer best represents the complex and non-linear relationship between the predisposing factors and landslide occurrence in the study area.

The susceptibility model developed in this study is based on a single-event inventory with one extreme rainfall pattern. Slope failures are caused by the weakening of soil unsaturated shear strength as the soil becomes saturated due to rainfall infiltration. The destabilizing force exerted on the soil layer is related to the layer thickness and geotechnical properties as these factors affect normal stress and shear strength, respectively. Rainfall patterns and soil permeability dictate the rate of water infiltration into the soil; hence, both affect the saturation of the soil layer at a particular time and location. For example, if a soil has a shallow depth and a large permeability coefficient, it will be more sensitive to rainfall patterns with intensive rainfall over a short period. In contrast, if the soil layer is deep and the permeability is relatively small, rainfall patterns with low rainfall intensity over long periods of time will have greater effects on landslide occurrence. Therefore, in order to enhance the performance of the susceptibility model, a future follow-up study should be conducted using an updated multi-temporal landslide inventory generated with consideration of other rainfall conditions.

## 6. Conclusions

This study demonstrates the systematic procedure of determining the optimal structure of an ANN-based landslide susceptibility model for identifying landslide-prone areas in Mount Umyoen, Korea. The main objective of this study was to design the optimal structure of the proposed MLP model, taking into account the factor selection method and various non-linear activation functions. The seven main procedures to achieve this purpose were as follows: (1) collecting historical landslide data, (2) constructing landslide predisposing factors, (3) preparing training and validation datasets, (4) applying a factor selection to select suitable database subsets, (5) developing landslide susceptibility models, (6) validating and comparing landslide susceptibility models, and (7) selecting the best performing model.

The best model was the MLP model consisting of an $11 \times 5 \times 1$ structure with the hyperbolic tangent sigmoid function in the hidden layer and the logistic sigmoid function in the output layer. The validation process confirmed that the best model ($11 \times 5$ for the tan-sig function $\times 1$ for the log-sig function) had a sensitivity of 82.61%, specificity of 78.26%, accuracy of 80.43%, positive predictive value of 79.17%, negative predictive value of 81.82%, and an AUC value of 0.879. In addition, the Kappa index was 0.609, indicating substantial agreement between the observed and predicted values. As a final conclusion, the results of this study may be useful for preemptive response in landslide-risk areas.

**Author Contributions:** Conceptualization, D.-H.L.; Methodology, D.-H.L.; Software, D.-H.L.; Validation, D.-H.L.; Formal Analysis, D.-H.L.; Investigation, D.-H.L.; Resources, D.-H.L.; Data Curation, D.-H.L.; Writing-Original Draft Preparation, D.-H.L.; Writing-Review & Editing, Y.-T.K. and S.-R.L.; Visualization, D.-H.L.; Supervision, S.-R.L.; Project Administration, Y.-T.K. and S.-R.L.; Funding Acquisition, S.-R.L. All authors have read and agreed to the published version of the manuscript.

**Funding:** This research was supported by the Basic Research Laboratory Program through the National Research Foundation of Korea funded by the Ministry of Science and ICT (NRF-2018R1A4A1025765) and the Korea Ministry of Land, Infrastructure and Transport (MOLIT) as Innovative Talent Education Program for Smart City.

**Acknowledgments:** We would like to thank the editors of remote sensing and three anonymous reviewers who gave us valuable comments that would be of great help in improving the quality of our paper.

**Conflicts of Interest:** The authors declare no conflict of interest.

## Appendix A

**Table A1.** Frequency ratio between landslides and predisposing factors.

| Factor | Class | No. of Pixels in Domain | % of Pixels in Domain | No. of Landslide | % of Landslide | Frequency Ratio [1] |
|---|---|---|---|---|---|---|
| Elevation | 23.4–68.3 | 40,715 | 0.17 | 14 | 0.09 | 0.54 |
| (m) | 68.3–113 | 88,856 | 0.37 | 36 | 0.24 | 0.64 |
| | 113–158 | 57,120 | 0.24 | 51 | 0.34 | 1.41 |
| | 158–203 | 30,307 | 0.13 | 24 | 0.16 | 1.25 |
| | 203–248 | 16,333 | 0.07 | 26 | 0.17 | 2.52 |
| | 248–293 | 5949 | 0.02 | 0 | 0.00 | 0.00 |
| Slope | 0–12 | 36,935 | 0.15 | 0 | 0.00 | 0.00 |
| (degree) | 12–24 | 129,945 | 0.54 | 38 | 0.25 | 0.46 |
| | 24–36 | 64,684 | 0.27 | 98 | 0.65 | 2.40 |
| | 36–47 | 7479 | 0.03 | 15 | 0.10 | 3.18 |
| | 47–59 | 237 | 0.00 | 0 | 0.00 | 0.00 |
| Aspect | Flat | 3713 | 0.02 | 0 | 0.00 | 0.00 |
| | N | 29,017 | 0.12 | 11 | 0.07 | 0.60 |
| | NE | 28,588 | 0.12 | 19 | 0.13 | 1.05 |
| | E | 31,456 | 0.13 | 24 | 0.16 | 1.21 |
| | SE | 32,258 | 0.13 | 14 | 0.09 | 0.69 |
| | S | 32,651 | 0.14 | 24 | 0.16 | 1.16 |
| | SW | 29,854 | 0.12 | 21 | 0.14 | 1.11 |
| | W | 23,161 | 0.10 | 21 | 0.14 | 1.44 |
| | NW | 28,582 | 0.12 | 17 | 0.11 | 0.94 |
| Curvature | Concave | 112,264 | 0.47 | 98 | 0.65 | 1.38 |
| | Planar | 10,893 | 0.05 | 0 | 0.00 | 0.00 |
| | Convex | 116,123 | 0.49 | 53 | 0.35 | 0.72 |
| TRI | 0–12 | 6751 | 0.03 | 0 | 0.00 | 0.00 |
| | 12–24 | 54,262 | 0.23 | 5 | 0.03 | 0.15 |
| | 24–32 | 63,716 | 0.27 | 22 | 0.15 | 0.55 |
| | 32–42 | 59,885 | 0.25 | 45 | 0.30 | 1.19 |
| | 42–59 | 48,690 | 0.20 | 67 | 0.44 | 2.18 |
| | 59–95 | 5976 | 0.02 | 12 | 0.08 | 3.18 |
| SRR | 0–0.23 | 5230 | 0.02 | 0 | 0.00 | 0.00 |
| | 0.23–0.46 | 43,224 | 0.18 | 26 | 0.17 | 0.95 |
| | 0.46–0.53 | 130,487 | 0.55 | 92 | 0.61 | 1.12 |
| | 0.53–0.67 | 52,451 | 0.22 | 33 | 0.22 | 1.00 |
| | 0.67–0.89 | 7888 | 0.03 | 0 | 0.00 | 0.00 |
| SEI | −58−−36 | 979 | 0.00 | 0 | 0.00 | 0.00 |
| | −36−−14 | 46,387 | 0.19 | 41 | 0.27 | 1.40 |
| | −14–7.5 | 104,677 | 0.44 | 44 | 0.29 | 0.67 |
| | 7.5–29 | 82,189 | 0.34 | 54 | 0.36 | 1.04 |
| | 29–51 | 5048 | 0.02 | 12 | 0.08 | 3.77 |
| TWI | 1.2–2.3 | 69,465 | 0.29 | 44 | 0.29 | 1.00 |
| | 2.3–2.8 | 85,422 | 0.36 | 62 | 0.41 | 1.15 |
| | 2.8–3.5 | 59,651 | 0.25 | 37 | 0.25 | 0.98 |
| | 3.5–4.6 | 18,901 | 0.08 | 8 | 0.05 | 0.67 |
| | 4.6–9.9 | 5841 | 0.02 | 0 | 0.00 | 0.00 |
| STI | 0–5 | 28,743 | 0.12 | 0 | 0.00 | 0.00 |
| | 5–25 | 129,887 | 0.54 | 55 | 0.36 | 0.67 |
| | 25–50 | 55,727 | 0.23 | 64 | 0.42 | 1.82 |
| | 50–100 | 17,817 | 0.07 | 23 | 0.15 | 2.05 |
| | >100 | 7106 | 0.03 | 9 | 0.06 | 2.01 |
| SPI | 0–5 | 13,142 | 0.05 | 0 | 0.00 | 0.00 |
| | 5–50 | 103,820 | 0.43 | 34 | 0.23 | 0.52 |
| | 50–100 | 50,676 | 0.21 | 43 | 0.28 | 1.34 |
| | 100–500 | 58,628 | 0.25 | 63 | 0.42 | 1.70 |
| | >500 | 13,014 | 0.05 | 11 | 0.07 | 1.34 |
| Distance | 0–21 | 79,705 | 0.33 | 46 | 0.30 | 0.91 |
| from | 21–43 | 73,801 | 0.31 | 45 | 0.30 | 0.97 |
| stream | 43–67 | 57,808 | 0.24 | 44 | 0.29 | 1.21 |
| (m) | 67–120 | 26,875 | 0.11 | 16 | 0.11 | 0.94 |
| | 120–250 | 1091 | 0.00 | 0 | 0.00 | 0.00 |

**Table A1.** *Cont.*

| Factor | Class | No. of Pixels in Domain | % of Pixels in Domain | No. of Landslide | % of Landslide | Frequency Ratio [1] |
|---|---|---|---|---|---|---|
| Lithology | Metamorphic | 179,607 | 0.75 | 142 | 0.94 | 1.25 |
| | Sedimentary | 57,344 | 0.24 | 9 | 0.06 | 0.25 |
| | No data | 2329 | 0.01 | 0 | 0.00 | 0.00 |
| Weathering | High | 109,136 | 0.46 | 105 | 0.70 | 1.52 |
| | Moderate | 127,815 | 0.53 | 46 | 0.30 | 0.57 |
| | No data | 2329 | 0.01 | 0 | 0.00 | 0.00 |
| Effective soil depth (cm) | 1–33 | 90,774 | 0.39 | 100 | 0.66 | 1.70 |
| | 33–49 | 56,426 | 0.24 | 31 | 0.21 | 0.87 |
| | 49–56 | 68,307 | 0.29 | 20 | 0.13 | 0.46 |
| | 56–69 | 21,444 | 0.09 | 0 | 0.00 | 0.00 |
| | No data | 2,329 | 0.01 | 0 | 0.00 | 0.00 |
| Soil texture | Silty loam | 90,712 | 0.38 | 41 | 0.27 | 0.72 |
| | Sandy loam | 146,239 | 0.61 | 110 | 0.73 | 1.19 |
| | No data | 2329 | 0.01 | 0 | 0.00 | 0.00 |
| Soil type [2] | $B_1$ | 117,481 | 0.49 | 109 | 0.72 | 1.47 |
| | $B_2$ | 107,142 | 0.45 | 42 | 0.28 | 0.62 |
| | $B_3$ | 12,328 | 0.05 | 0 | 0.00 | 0.00 |
| | No data | 2329 | 0.01 | 0 | 0.00 | 0.00 |
| Soil density | Medium dense | 27,019 | 0.11 | 23 | 0.15 | 1.35 |
| | Loose | 188,702 | 0.79 | 116 | 0.77 | 0.97 |
| | Very loose | 21,230 | 0.09 | 12 | 0.08 | 0.90 |
| | No data | 2329 | 0.01 | 0 | 0.00 | 0.00 |
| Forest type | Coniferous | 5796 | 0.02 | 1 | 0.01 | 0.27 |
| | Broadleaf | 216,285 | 0.90 | 140 | 0.93 | 1.03 |
| | Mixed | 14,914 | 0.06 | 9 | 0.06 | 0.96 |
| | No forest | 2285 | 0.01 | 1 | 0.01 | 0.69 |
| Forest Density | Dense | 226,644 | 0.95 | 146 | 0.97 | 1.02 |
| | Moderate | 10,351 | 0.04 | 4 | 0.03 | 0.61 |
| | No forest | 2285 | 0.01 | 1 | 0.01 | 0.69 |
| Distance from road (m) | 0–130 | 100,938 | 0.42 | 59 | 0.39 | 0.93 |
| | 130–300 | 59,632 | 0.25 | 43 | 0.28 | 1.14 |
| | 300–550 | 43,386 | 0.18 | 23 | 0.15 | 0.84 |
| | 550–900 | 17,734 | 0.07 | 13 | 0.09 | 1.16 |
| | 900–1500 | 17,590 | 0.07 | 13 | 0.09 | 1.17 |

[1] Frequency ration is calculated by % of pixels in domain / % of landslide. [2] $B_1$ is dry brown forest soil, $B_2$ is slight dry brown forest soil, and $B_3$ is moderately moist brown forest soil.

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
