# Peer review of "Shallow Landslide Susceptibility Models Based on Artificial Neural Networks Considering the Factor Selection Method and Various Non-Linear Activation Functions"

_remotesensing, doi:10.3390/rs12071194_

Round 1

Reviewer 1 Report

Dear Authors, I have read your manuscript "Shallow Landslide Susceptibility Models Based on Artificial Neural Networks Considering the Factor Selection method and Various Non-Linear Activation Functions" and found it good, clear, well written and worth of being published after some modifications.

Please, refer to the attached document and all my comments therein. 

Regards. 

Author Response

We thank the editor and reviewers for their insightful comments which truly helped enrich the manuscript. In the revised manuscript, we have addressed all the comments provided by the three reviewers.

For the comments raised by the reviewers, we have provided the followings:

a - Point by point responses to comments and

b - Revised text is marked in blue.

Reviewer 2 Report

In this work, the Authors show an application of artificial neural network (ANN) algorithm involving the factor selection method and various non-linear activation functions in order to produce the landslide susceptibility map of Mt. Umyeon in Korea. In order to this, a total of 151 historical landslide events and 20 influencing factors consisting of GIS-based morphological, hydrological, geological, and land cover datasets were used. Furthermore, the performance of the models was evaluated using the receiver operating characteristic curve, Kappa index, and five statistical indices (sensitivity, specificity, accuracy, positive predictive value, negative predictive value) with the training dataset.

The study provides an accurate description of the used approach and of the analyses carried out with a simple and clear structure.

It is an original contribution, appropriate material for Remote Sensing journal. For these reasons, I suggest publishing the paper after minor revision. 

Comments:

Pg.4 line 128: The resolution of the DEM used is unclear. The authors talk about 1:25000 scale but then they use 5x5m resolutions for the different themes. Explain this point better.

Pg.6 Geological Types Section: It might be useful to insert a geological map

Pg.6 line 192: How were the thickness values obtained?

Pg.6 lines 193-195: This statement is not entirely true, it depends on the lithology in particular on permeability. Please, rewrite the sentence.

Pg.9: Some parameters are evidently not significant, could they be removed from the elaborations?

Pg.10: Equation (7): please insert the reference

Pg.10 line 233: What algorithm was used for the random selection?

Pg.17 line 445: please correct the figure reference (Fig. 6)

Author Response

(The authors gave the same response as above.)

Reviewer 3 Report

The manuscript “Shallow Landslide Susceptibility Models Based on Artificial Neural Networks Considering the Factor Selection method and Various Non-Linear Activation Functions” presents a methodology for objectively evaluate performances of Artificial Neural Networks in producing Landslide Susceptibility Models.

The manuscript is well structured and steps of the analysis are clearly exposed. The English language is of good quality. In my opinion the manuscript should be published after a (very) minor edit: the authors refer to “internal” and “external” factors that rule landslides occurrences. In literature, also in one of the papers that is cited in the manuscript (Reichenbach et al.,2018), these are identified as “predisposing”  and “triggering” factors. In addition, I would ask the authors to replace, throughout all the manuscript, “influencing” into “predisposing” factors.

Author Response

(The authors gave the same response as above.)

Round 2

Reviewer 1 Report

Dear Authors, just a few remarks before final acceptance.

  • references 11-12 i suggested should be used at line 69-70, to show how expensive and time consuming is providing geotechnical measures for the application of physically based models over large areas.
  •   lines 601-611. My remark in the first round of review was maybe misunderstood. Susceptibility maps are independent from dynamic factors like rainfall. Hence, a good susceptibilty map is perfomed with multi-temporal landslide inventories that are representative of the rainfall conditions found in the study area. If you use a single-event inventory this condition is not met. To sum up, in the future the susceptibility assessment should be updated including in the calibration other landslides triggered by other rainfall conditions.  

Author Response

We appreciate your valuable comments and suggestions. The overall manuscript has been carefully revised as you recommended.
